# OpenReview forum: "Dynamic-SUPERB Phase-2: A Collaboratively Expanding Benchmark for Measuring the Capabilities of Spoken Language Models with 180 Tasks"
_ICLR.cc/2025/Conference — ICLR 2025 Poster_

### Official Review · Reviewer_j6zR · 2024-11-03

**Soundness:** 3
**Presentation:** 4
**Contribution:** 3
**Rating:** 8
**Confidence:** 3

**Summary:**

The paper suggests a new (milestone of a) benchmark to test the audio analysis tasks of universal speech models. It presents an overview of 180 tasks, contributed by the research community and categorized into a a taxonomy, which all have in common that one or multiple audios, as well as a task prompt are presented to a to-be-evaluated model, which then provides a language response, aiming to solve the given task. The response is then evaluated by a post-processing pipeline, potentially including a "judge LLM". Besides providing the evaluation benchmark pipeline, the authors conducted an analysis of current universal (at times specialised) speech models. They conclude that no model excels at all tasks, but strengths and weaknesses can be observed across different models and tasks.

**Strengths:**

The paper addresses an important point challenging the "universality" claim universal speech models. The authors claim to have gathered the largest benchmark by a large margin, improving on top of presumably their own work and provide a detailed analysis comparing several current speech models. The benchmark has the potential to have a large impact on the research community and it provides useful insights into the limitations of LLMs.

**Weaknesses:**

L.371: Maybe it is worth that the authors iterate why they didn't choose a more "direct" approach for the evaluation, in which the model is prompted to give a single word based on the fixed set of possible answers. In the case of emotion recognition, it could be "answer with a single word out of the choices 'happy, sad, neutral,...'", which would allow a more direct evaluation. Especially in the case of emotion recognition a non-pre-defined set of emotions will also make it difficult for the judge LLM (and humans for reference) to judge how accurate the result is. This might at least be considered as a second evaluation metric, this seems more or less as what is being done for regression, but why isn't this "extra" step directly asked from the model being evaluated? For sequences evaluation it is stated that the evaluation of unprocessed outputs "guarantees consistency and objectivity".

Section 5.1. It might be difficult to define for some tasks, but a comparison to chance level and/or state-of-the-art task-specific models and/or "perfect" scores would seem a very interesting insight. As is, it seems that the evaluation allows relative comparison between -quite elaborate- models, which allows to see which model is better amongst two, but doesn't really allow to see if the models are still doing reasonably well in these tasks and how much room for improvement there still is. For instance in the case of of spatial audio, none of the models are performing better than the baseline, but it not clear, if none of them or all of them are reasonably good at the task. This is somewhat done in section 5.2. but seems to be missing in 5.1.

**Questions:**

What is the justification of using Whisper + LLaMA as the baseline? There should be some explanation why this particular architecture has been chosen.

L.267: I think the phrase for distinguishing sources "speech is produced by humans" could be rephrased, as humans could also "produce" other non-speech sounds. Speech should rather be connected to the vocal chords.
L.286: The example of Speech Enhancement to mitigate does not become clear. Above it is stated that the models should produce text as output (which can maybe be interpreted as lists). Under Speech Enhancement for mitigating noise (unless you mean only e..g, SNR estimation) I would expect an audio output.

**Details Of Ethics Concerns:**

With the amount of tasks and datasets presented here, it seems important to verify that usage of all datasets is allowed. However, there do not seem to be any unethical considerations in the types of selected datasets.

---

> ### Author Response · Authors · 2024-11-24
> **Author Response to Reviewer j6zR (1/2)**
>
> Dear Reviewer j6zR,
>
> Thank you for your thoughtful and encouraging feedback on our paper. Below, we address the weaknesses and questions raised in your review.
>
> > Weakness 1: Maybe it is worth that the authors iterate why they didn't choose a more "direct" approach for the evaluation, in which the model is prompted to give a single word based on the fixed set of possible answers. In the case of emotion recognition, it could be "answer with a single word out of the choices 'happy, sad, neutral,...'", which would allow a more direct evaluation. Especially in the case of emotion recognition a non-pre-defined set of emotions will also make it difficult for the judge LLM (and humans for reference) to judge how accurate the result is. This might at least be considered as a second evaluation metric, this seems more or less as what is being done for regression, but why isn't this "extra" step directly asked from the model being evaluated? For sequences evaluation it is stated that the evaluation of unprocessed outputs "guarantees consistency and objectivity".
>
> Thank you for the suggestion. Here are the reasons we do not use the “direct” approach.
>
> In Phase 2, not all instructions included all available options. Our goal was to create task instructions (prompts) that are diverse and closely resemble actual human usage. Therefore, we did not strictly require task contributors to list all possible options. We believe that omitting options more accurately reflects real-world usage. For instance, when a user interacts with a spoken LLM for emotion recognition, they might directly ask, “What's the speaker’s emotion?” instead of listing all options, such as, “What’s the speaker’s emotion? The answer could be happy, sad, angry, or neutral.”
>
> In addition, in Phase-1 of Dynamic SUPERB, all available options were included in the instructions (prompts) for classification tasks. However, it was observed that several models struggled to follow this kind of instruction. For example, in an emotion recognition task with a list of options, a model might output, “The answer could be happy,” instead of a single word “happy,” even when explicitly asked to respond with a single word. This made it difficult to apply direct evaluation methods such as Exact Match or word search (e.g., simply checking if the answer is present in the response). These methods often resulted in scores of zero or worse-than-random performance for the currently available models, even though these models can handle certain speech tasks to some extent. This is one reason we do not evaluate models using the “direct” approach, as such evaluations may be highly related to instruction-following capabilities. Currently, Dynamic SUPERB Phase-2 emphasizes how well a spoken language model can understand speech and sounds. Of course, instruction-following capability remains crucial. In the future, we may include more tasks that better examine the instruction-following capabilities of spoken LMs.
>
> For sequence generation tasks such as ASR, we directly used unprocessed output because, without listening to the actual audio input, even humans find it difficult to determine the correct processing. For example, if a speaker says, “That man said he could scoop these things in three red bags...,” and a spoken language model performing ASR outputs, “Sure, that man said he could scoop...,” then, without access to the audio input, it would not be easy for a post-processing LLM or human to determine whether “sure” or “sure, that man said” is content unrelated to the transcription. Therefore, in Phase-2, we simply use unprocessed output for these tasks, and we are currently working on developing new approaches to better evaluate these tasks in Phase-3.
>
>
> > Weakness 2: It might be difficult to define for some tasks, but a comparison to chance level and/or state-of-the-art task-specific models and/or "perfect" scores would seem a very interesting insight. As is, it seems that the evaluation allows relative comparison between -quite elaborate- models, which allows to see which model is better amongst two, but doesn't really allow to see if the models are still doing reasonably well in these tasks and how much room for improvement there still is. For instance in the case of of spatial audio, none of the models are performing better than the baseline, but it not clear, if none of them or all of them are reasonably good at the task. This is somewhat done in section 5.2. but seems to be missing in 5.1.
>
> Thank you for bringing this to our attention. We used relative scores to report performance for simplified comparison in the main text and provided the full, complete performance scores in Appendix C. To better reflect task difficulty, we will introduce an additional random baseline to enable benchmark users to better understand how well these models are performing.

---

> ### Author Response · Authors · 2024-11-24
> **Author Response to Reviewer j6zR (2/2)**
>
> > Question 1: What is the justification of using Whisper + LLaMA as the baseline? There should be some explanation why this particular architecture has been chosen.
>
> Whisper-LLaMA, as described in this paper, is a cascade combining a widely used publicly available ASR model (Whisper) with a widely used publicly available text LLM (LLaMA). It directly leverages the strong NLP capabilities of LLaMA by transcribing speech into text using the off-the-shelf ASR model, Whisper. We use it as a baseline because of its cost-effectiveness in building a model capable of taking speech as input and producing text as output.
>
> > Question 2: I think the phrase for distinguishing sources "speech is produced by humans" could be rephrased, as humans could also "produce" other non-speech sounds. Speech should rather be connected to the vocal chords. L.286: The example of Speech Enhancement to mitigate does not become clear. Above it is stated that the models should produce text as output (which can maybe be interpreted as lists). Under Speech Enhancement for mitigating noise (unless you mean only e..g, SNR estimation) I would expect an audio output.
>
> Thank you for pointing this out. We will modify the statement to be more precise ("Speech is produced by human vocal cords."). In line 286, we will explicitly mention that currently we only have understanding tasks in the Speech Enhancement track to avoid confusion, and we plan to include speech/music/audio generation tasks in Phase-3.
>
> We hope these explanations help clarify your concerns.
>
> Sincerely,
>
> Authors of Submission 6527

---

### Official Review · Reviewer_59hW · 2024-11-04

**Soundness:** 3
**Presentation:** 3
**Contribution:** 3
**Rating:** 8
**Confidence:** 3

**Summary:**

This paper introduces a "Dynamic-SUPERB Phase-2", a benchmark for evaluating of audio-in-capable LLMs. This benchmark contains 180 tasks that cover a a large variety of tasks, spanning from music and audio classification to speech recognition, nonce word detection, intent and age classification, etc. These tasks were collected collaboratively. The audio-LLM are assumed to take in text instruction, an audio snippet, and produce text output. The interaction with the model is performed solely by the instruction, no finetuning is assumed. Depending on the task, the output encodes either classification label, regression value, or a natural language string. The benchmark is split into (a) full version, (b) an easier to operate representative sub-sample of tasks.

Given the benchmark, the authors also run a study comparing several off-the-shelf audio-capable LLMs, such as AudioQWEN, SALMONN, WhisperLLAMA, etc. One particular issue encountered is that various audio-LLMs would produce outputs in different formats, so the authors leverage another textual LLM to "translate/interpret" their outputs to the desired format.

**Strengths:**

* The paper tackles perhaps one of the most important problems in the audio-LLM space: the lack of audio-text benchmarks. In contrast to text-LLMs, where sets of hundreds of tasks are readily available, audio-LLM development is hindered by the lack of standard testing suites.
* I find the organization of the benchmarks to be thoroughly thought through.
* The paper runs a nice study on comparing off-the-shelf models, thus validating the proposed benchmarks.
* The paper validates referee model selection against human preferences.

**Weaknesses:**

* Using GPT-4o as a referee, I believe, jeopardizes reproducibility of the reported results and usefulness of the paper. Imagine that in two years someone wants to compare their model to the benchmark numbers reported in this paper. Would GPT-4o still be around? Would this particular version of it be still accessible? This is even more puzzling, since, according to Appendix E, LLaMA-3.1-70B-Instruct is extremely close to GPT-4o here. It seems to me that using an openly accessible model here would be extremely beneficial.

* Regarding the model evaluation study: it is not clear how sensitive the tested models are wrt the prompts used. It would be very nice to have a small study on that.

* Whenever the evaluated model doesn't follow the instruction, its score is multiplicatively penalized by (1-N/A) or 1.0/(1-N/A) (depending on the metric). This is convenient as allows to have a single number but it totally hides how good models are at instruction-following.

**Questions:**

* The paper text reports that In Figure 3, Whisper-LLaMa performs best in speech recognition and understanding. However, when we look at Table 2, it is way behind SALMONN in phone recognition. Looking at ASR performances across tables, it does seem that SALMONN mostly falls behind on a few non-English transcriptions (SUPERB OOD Asr Zh). I wonder if there is a more nuanced story here than is told in S5.1.

* Table 7 has inconsistent WER reporting, ie "Children Song Transcript" has WER not scaled by 100% while it is scaled in other places.

* It would be nice to have the "N/A rate" reported for the models.

---

> ### Author Response · Authors · 2024-11-24
> **Author Response to Reviewer 59hW (1/2)**
>
> Dear Reviewer 59hW,
>
> Thank you for your thoughtful and encouraging feedback on our paper. Below, we address the weaknesses and questions raised in your review.
>
> > Weakness 1: Using GPT-4o as a referee, I believe, jeopardizes reproducibility of the reported results and usefulness of the paper. Imagine that in two years someone wants to compare their model to the benchmark numbers reported in this paper. Would GPT-4o still be around? Would this particular version of it be still accessible? This is even more puzzling, since, according to Appendix E, LLaMA-3.1-70B-Instruct is extremely close to GPT-4o here. It seems to me that using an openly accessible model here would be extremely beneficial.
>
> Thank you for pointing this out. We have conducted a comprehensive evaluation using open-source LLMs to enhance reproducibility. Initially, we did not include them in the main text due to space limitations and put the correlation results in the Appendix. However, since the evaluation results of open-source models are more critical, we will include them in the main text in the subsequent revisions. Additionally, we will explicitly encourage benchmark users to leverage these open-source models on our GitHub repository and website (we are not able to directly share the link here because of the anonymity).
>
> > Weakness 2: Regarding the model evaluation study: it is not clear how sensitive the tested models are wrt the prompts used. It would be very nice to have a small study on that.
>
> We have conducted a preliminary experiment on 15 classification tasks in Dynamic-SUPERB. For each task, we first calculated the average performance across all evaluated models for each type of instruction (prompt) and then computed the variance across different instructions. We observed that the variance across tasks ranges from very small values (e.g., MARBLE tasks) to larger values (e.g., SUPERB tasks); however, these variances remain much smaller compared to the performance range of the evaluated baseline models. It is worth noting that it is difficult to directly assess prompt sensitivity because performance is also influenced by the speech/audio input. Furthermore, to prevent models from memorizing specific instructions for a task (e.g., only recognizing "transcribe the speech" for ASR), we have encouraged task contributors to provide instructions as diverse as possible for each task. Consequently, a strong instruction comprehension capability is necessary for a model to perform sufficiently well on a task.
>
> | Task | Prompt Variance | Model Acc. Range (max - min) |
> | ---------- | :----------: | :----------: |
> | SUPERB Intent Classification | 0.0197 | 0.5550 |
> | SUPERB Keyword Spotting | 0.0281 | 0.5950 |
> | SUPERB Query by Example | 0.0326 | 0.5100 |
> | L2-English Prosodic Ranking | 0.0141 | 0.2611 |
> | L2-English Accuracy Ranking | 0.0392 | 0.2528 |
> | Prosody Naturalness (Lexical) | 0.0123 | 0.4942 |
> | Prosody Naturalness (Protosyntax) | 0.0177 | 0.4924 |
> | Animal Classification | 0.0126 | 0.7150 |
> | Age Classification | 0.0050 | 0.3500 |
> | Human Screaming Detection | 0.0228 | 0.6250 |
> | Cornell Birdcall Identification | 0.0024 | 0.1333 |
> | Instrument Classification (Nsynth) | 0.0077 | 0.5318 |
> | Instrument Pitch Classification | 0.0010 | 0.1800 |
> | MARBLE Key Detection | 0.0017 | 0.1700 |
> | MARBLE Vocal Technique Detection | 0.0035 | 0.1450 |
>
> > Weakness 3: Whenever the evaluated model doesn't follow the instruction, its score is multiplicatively penalized by (1-N/A) or 1.0/(1-N/A) (depending on the metric). This is convenient as allows to have a single number but it totally hides how good models are at instruction-following.
>
> We reported the scaled performance in the main text due to limited space. For a more comprehensive evaluation, we provide the N/A rate for each model in each task in Appendix C.

---

> ### Author Response · Authors · 2024-11-24
> **Author Response to Reviewer 59hW (2/2)**
>
> > Question 1:The paper text reports that In Figure 3, Whisper-LLaMa performs best in speech recognition and understanding. However, when we look at Table 2, it is way behind SALMONN in phone recognition. Looking at ASR performances across tables, it does seem that SALMONN mostly falls behind on a few non-English transcriptions (SUPERB OOD Asr Zh). I wonder if there is a more nuanced story here than is told in S5.1.
>
> Whisper-LLaMA performed worse in phoneme recognition probably because Whisper provides word-level transcriptions, and LLaMA may not be able to convert words (graphemes) to phonemes. In contrast, SALMONN is explicitly trained for phoneme recognition, as mentioned in their paper. For multilingual ASR, we noticed that SALMONN falls behind because it sometimes performs translation of the utterances instead of simply doing ASR. Overall, SALMONN still performs reasonably well on several tasks. We will also try to include these discussions in the results section (Sec. 5.1) of the paper.
>
> > Question 2: Table 7 has inconsistent WER reporting, ie "Children Song Transcript" has WER not scaled by 100% while it is scaled in other places.
>
> Thank you for bringing this to our attention. We will correct it.
>
> > Question 3: It would be nice to have the "N/A rate" reported for the models.
>
> In Appendix C, we report the N/A rate for each model in each task. However, due to space limitations, we may not be able to include them in the main text. We will update the complete information on our website to ensure easy and clear access.
>
> We hope these explanations help clarify your concerns.
>
> Sincerely,
>
> Authors of Submission 6527

---

> > ### Comment · Reviewer_59hW · 2024-11-24
> >
> > Thanks a lot for the reply!

---

### Official Review · Reviewer_wDFb · 2024-11-04

**Soundness:** 3
**Presentation:** 3
**Contribution:** 3
**Rating:** 6
**Confidence:** 3

**Summary:**

This paper presents Dynamic-SUPERB Phase-2, a benchmark designed to evaluate the capabilities of instruction-based universal speech models. By expanding the task count of its predecessor to 180, it becomes the largest benchmark for speech, music, and audio, encompassing a diverse array of tasks contributed by the global research community. This work is poised to drive further advancements in developing universal spoken language models and their evaluation methodologies.

**Strengths:**

1. Dynamic-SUPERB Phase-2 is the largest and most comprehensive benchmark for instruction-based universal speech models. It encompasses a wide range of tasks across speech, music, and audio, all paired with natural language instructions to evaluate models' cross-modal instruction-following abilities. This is a well-motivated and forward-looking approach that aligns with future trends in universal speech models, offering strong guidance for the field.
2. The paper is clearly written and easy to follow.

**Weaknesses:**

1. The benchmark lacks tasks for audio, speech, or music generation.
2. A primary concern is that the evaluation metrics heavily rely on large language models (LLMs) as referees. The reliability of these metrics is highly dependent on the capabilities of the LLMs themselves, which affects the benchmark's robustness and comparability. This reliance may also limit the benchmark's ability to expand to more complex tasks.

**Questions:**

My questions are listed above.

---

> ### Author Response · Authors · 2024-11-24
> **Author Response to Reviewer wDFb**
>
> Dear Reviewer wDFb,
>
> Thank you for your thoughtful and encouraging feedback on our paper. Below, we address the weaknesses raised in your review.
>
> > Weakness 1: The benchmark lacks tasks for audio, speech, or music generation.
>
> We acknowledge that the benchmark currently does not include these types of generation tasks (hereafter referred to as audio generation tasks). During the call-for-task phase, we did receive some submissions related to audio generation tasks. However, we observed that there are very few publicly available models capable of handling a wide range of audio generation tasks. As a result, these tasks were not included in Phase-2. Moving forward, we plan to collect and organize more audio generation tasks in Phase-3 to make the benchmark even more comprehensive.
>
> > Weakness 2: A primary concern is that the evaluation metrics heavily rely on large language models (LLMs) as referees. The reliability of these metrics is highly dependent on the capabilities of the LLMs themselves, which affects the benchmark's robustness and comparability. This reliance may also limit the benchmark's ability to expand to more complex tasks.
>
> Regarding the LLM evaluation, we have tested the correlation between LLMs’ judgments and human annotations in Appendix E (Tables 11 and 12), where GPT-4o and LLaMA-3.1-70B-Instruct show high alignment. For classification tasks, GPT-4o and LLaMA-3.1-70B-Instruct achieved 98.67% and 96.67% accuracy on the human-labeled subset respectively. On the other hand, for regression tasks, GPT-4o and LLaMA-3.1-70B-Instruct had 92% and 90% accuracy. Therefore, we believe it is safe to say that LLM evaluations are trustworthy and usable for the tasks in the benchmark for now.
> For more complex tasks that might be integrated into the benchmark in the future, we will ask task contributors (and ourselves) to provide a preliminary investigation comparing LLM evaluations and human annotations. We will also open-source our inference pipeline and present the evaluation results on the Dynamic-SUPERB website.
>
> We hope these explanations help clarify your concerns.
>
> Sincerely,
>
> Authors of Submission 6527

---

### Official Review · Reviewer_59Pj · 2024-11-07

**Soundness:** 3
**Presentation:** 3
**Contribution:** 2
**Rating:** 6
**Confidence:** 4

**Summary:**

This paper builds on the prior work, Dynamic-Superb, by expanding its scope as an audio LLM benchmark to include a broader range of tasks centered on audio and speech understanding. It includes diverse tasks across speech, music, and general audio comprehension. The authors also evaluate several well-known audio LLMs, including SALMONN and Qwen-Audio, on this benchmark, showing that none of the current open-source audio LLMs can yet achieve satisfactory performance across all tasks.

**Strengths:**

This work is comprehensive and well presented. It brings the largest benchmark by far for speech and audio evaluation. It also conducts detailed experiments to assess the performance of several popular audio LLMs on the proposed benchmark.

**Weaknesses:**

There're a few possible issues that may further improve this paper.
- What is the motivation of creating this new benchmark? How will it guide the research community in advancing audio LLM research? I think the authors could emphasize this point further.
- I understand that the number of tasks has significantly expanded in Phase 2 compared to Phase 1. However, what is the primary focus of these additional tasks? Are there specific challenges unresolved in Phase 1 that Phase 2 addresses?

**Questions:**

Please refer to the weakness part.

---

> ### Author Response · Authors · 2024-11-24
> **Author Response to Reviewer 59Pj**
>
> Dear Reviewer 59Pj,
>
> Thank you for your thoughtful and encouraging feedback on our paper. Below, we address the weaknesses raised in your review.
>
> > Weakness 1: What is the motivation of creating this new benchmark? How will it guide the research community in advancing audio LLM research? I think the authors could emphasize this point further.
>
> Despite significant advancements in spoken LLMs, most research papers evaluate these models on a limited set of tasks. This narrow focus fails to determine whether spoken LLMs can universally comprehend all aspects of speech. Existing benchmarks like SUPERB have been widely used in research, but they mainly focus on building task-specific models from a foundation model and are confined to a relatively small set of tasks. Therefore, they are not suitable for comprehensively evaluating spoken language models. To address this gap, we propose Dynamic-SUPERB Phase-2 as a benchmark for evaluating spoken language models that can take both speech and text as inputs across a wide range of tasks (over a hundred). The ultimate goal of this benchmark is to advance the development of spoken language models towards handling speech tasks almost universally, similar to how LLMs function in NLP. Besides, no existing spoken LLM benchmark currently includes such a diverse range of tasks (Please refer to Table 1 in the paper).
>
>
> > Weakness 2: I understand that the number of tasks has significantly expanded in Phase 2 compared to Phase 1. However, what is the primary focus of these additional tasks? Are there specific challenges unresolved in Phase 1 that Phase 2 addresses?
>
> The main differences between Dynamic-SUPERB Phase-1 and Phase-2 are **(1) task coverage** and **(2) task taxonomy**. In Phase-1, the benchmark only covered 55 classification tasks with a simplified evaluation method (Exact Match). In Phase-2, we have expanded the benchmark to cover more diverse tasks, including sequence generation tasks such as ASR and speech translation. Importantly, the benchmark has been expanded through collaborative, community-driven activities, which help it align better with the community's common interests in different aspects of speech, audio, and music tasks (e.g., safety-related tasks). Furthermore, we have introduced new approaches to evaluate each task, utilizing LLMs to provide more reasonable and flexible solutions in addition to Exact Match. In Phase-1, tasks were categorized into six dimensions based on the specific speech attribute involved (semantic, paralinguistic, speaker, etc.). In Phase-2, we redesigned the task taxonomy according to the application or problem the task is intended to solve, making it much more fine-grained and closely aligned with the taxonomies used in speech conferences and journals.
>
> We hope these explanations help clarify your concerns.
>
> Sincerely,
>
> Authors of Submission 6527

---

### Author Response · Authors · 2024-11-24
**General Response to All Reviewers**

Dear Reviewers,

We sincerely appreciate your thoughtful and encouraging feedback on our paper. We are grateful that you recognize the significance of our work in presenting the largest and most comprehensive benchmark to date for evaluating audio and speech models. The breadth and diversity of the 180 tasks we included, which span speech, music, and general audio, were crucial to our goal, and we're pleased that this has been well received by all of you. In addition, we are delighted that you have collectively noted that our paper is well-written and clearly presented, making the complex content accessible and easy to follow. Furthermore, the collaborative effort in task collection from the global research community was also mentioned in all reviews, highlighting the importance of community involvement in advancing this field.

We are thankful for your recognition of these strengths and believe that your feedback will help us further refine and improve our work. Regarding the weaknesses and questions you've raised, we have replied to each reviewer individually. Thank you again for your valuable reviews and for contributing to the advancement of research in spoken language modeling.

Sincerely,

Authors of Submission 6527

---

### Meta-Review · Area_Chair_b9eJ · 2024-12-07

**Metareview:**

Dynamic-SUPERB Phase-2 is the largest and most comprehensive benchmark for instruction-based universal speech models, covering a wide range of tasks across speech, music, and audio with natural language instructions.  It tackles the significant problem of the lack of audio-text benchmarks in the audio-LLM space, which hinders development compared to text-LLMs. The benchmark has the potential to significantly impact the research community by providing useful insights into the limitations of LLMs and challenging the "universality" claim of universal speech models.

**Additional Comments On Reviewer Discussion:**

The reviewers raised several concerns: 1) using GPT-4o as a referee affects reproducibility and future accessibility, 2) model sensitivity to prompts and the current scoring method need improvement, 3) the motivation for the new benchmark and its guidance for audio LLM research require more emphasis, and 4) a more direct evaluation approach and comparisons to chance levels or state-of-the-art models would be beneficial. Most of these issues have been addressed during the author-reviewer discussion.

---

### Decision · Program_Chairs · 2025-01-22

Accept (Poster)